



# The evolution of warm rain in trade-wind cumulus during EUREC[4]A

Gary Lloyd[1,2], Alan Blyth[3], Zhiqiang Cui[3], Thomas Choularton[2], Keith Bower[2], Martin Gallagher[2], Michael Flynn[2], Nicholas Marsden[1,2], Leif Denby[3], Peter Gallimore[2]

[1]National Centre for Atmospheric Science (NCAS), University of Manchester, Manchester, M13 9PL, UK
[2]Centre for Atmospheric Science, University of Manchester, Manchester, M13 9PL, UK
[3]National Centre for Atmospheric Science (NCAS), University of Leeds, Leeds.

*Correspondence to*: Gary Lloyd (gary.lloyd@manchester.ac.uk)

**Abstract.**

In this paper measurements are presented of the observed properties of aerosols and microphysics of clouds associated with
the characteristics of precipitation in convective clouds that formed off the east coast of Barbados during EUREC[4]A. Most
data were gathered by the instrumented British Antarctic Survey Twin Otter aircraft supported by detailed in-situ aerosol
measurements at the Ragged Point observatory on Barbados as well as HALO and PoldiRad radars, dropsonde and satellite
data. The development of precipitation was studied in the three aerosol regimes previously reported, i.e. one low aerosol regime
and two containing desert dust that had been advected across the Atlantic Ocean. The later dust event also contained evidence
of biomass burning aerosol. Results showed that the maximum intensity of rain was similar for all the aerosol regimes. Clouds
that developed in an environment with high aerosol loading tended to be deeper than those that developed in the clean
environment. It was also found that the greatest intensities occurred in clouds that had aggregated, in agreement with previous
work.

**1 Introduction**

Shallow oceanic cumulus clouds in the trade-wind regions are a semi-permanent feature of the daily weather conditions and
one of the dominant cloud types in tropical and sub-tropical regions (Vial et al., 2019; Johnson et al., 1999). They comprise a
number of different macro-scale cloud forms that have been defined by names that describe their visual appearance and
organisation (e.g. sugar, gravel, fish and flowers) (Stevens et al., 2020; Bony et al., 2020; Schulz et al., 2021; Schulz et al.,
25  2022)

Previous research has contributed significant knowledge to our understanding of these cloud systems and dates back to the
Barbados Oceanographic and Meteorological Experiment (BOMEX) and the Atlantic Trade Wind Experiment (ATEX) in
1969 (Kuettner and Holland, 1969; Augstein et al., 1973) and more recently the Rain in Cumulus over the Ocean (RICO)
(Rauber et al., 2007) and the Cloud, Aerosol, Radiation and tuRbulence in the trade wInd regime over BArbados project
(CARRIBA) (Siebert et al., 2013).

The objectives of BOMEX mainly concerned the determination of the exchange of heat, moisture, and momentum between
the ocean to the atmosphere. In ATEX, measurements were made from three ships of the thermodynamics and kinematic
structure of the planetary boundary layer. The trade wind cumulus clouds and their environment were well observed during
both projects and they have provided important measurements for large-eddy model simulations of the trade-wind clouds and
their environment even to this day (e.g. Stevens et al., 2001; Siebesma et al., 2003; Heus et al., 2009; Eytan et al., 2022). No
cloud microphysics measurements were made during BOMEX or ATEX. However detailed microphysics measurements were
made in RICO and CARRIBA and the literature contains many detailed results concerning the development of warm rain and
the influence of aerosols and turbulence on the cloud drops. Seibert et al. (2013) for example presented findings from two
cases in CARRIBA with very different aerosol and cloud microphysical properties, but almost identical meteorological





conditions, which were similar to those observed during EUREC[4]A (discussed below). As expected, larger cloud droplets were observed in the case with lower aerosol concentrations. However there was no measurable difference in the properties of turbulent mixing and the influence on the cloud droplets.

Snodgrass et al. (2009) found from RICO observations from radar and aircraft in Barbuda that rainfall rates were closely tied to the type of mesoscale organization, with much of the rainfall originating from shallow cloud arcs associated with cold-pool outflows. They also determined that between 1.5% and 3.5% of the cloud area had rainfall rates greater than 1 mm/hr, and clouds with tops between about 3 and 4 km contributed about 50% of the total rainfall with most originating from arc shaped cumulus clusters (< 5km in height) linked to cold-pool outflows.


The motivation for previous projects and the work presented in this paper focus on the importance of the cumulus clouds in the trade wind region around Barbados and the difficulty they pose for models that need to parameterise clouds and atmospheric properties in such environments. Tropical cloud feedbacks have been identified as a key uncertainty in climate models (Bony and Dufresne, 2005) and the relationship between cloud microstructures and macroscale properties remains poorly understood

(Van Zanten, 2011). Any changes in the frequency of these relatively shallow clouds over the ocean has the potential to significantly impact the net solar radiation budget due to their relatively high albedo verses the ocean surface (approximately an order of magnitude net increase in reflected short wave radiation) and minimal impact on outgoing longwave radiation (Stevens and Feingold, 2009).

The intensity of precipitation was found by Radtke et al. (2022) to increase mainly with the size of convective cells, and the result was stronger when there were fewer cells, in a drier environment. They also found that the amount of precipitation was greater when there were many large cells and that the spatial organisation was less of a control. Radtke et al. (2023) performed ICON hectometer (hm)-scale simulations of the North Atlantic trades to determine if the development of precipitation is influenced by the spatial organisation of the clouds. Interestingly they found that the efficiency of precipitation formation was

reduced for stronger organisation. Evaporation below cloud base was also reduced. Their model results suggested that the reasons were that when the organisation was stronger, the updrafts were weaker, the cloud droplets were smaller and the environment was moister.

The evolution of the cloud droplet size distribution, and eventual production of precipitation is driven by a number of

different physical processes. Condensational growth occurs as cloud droplets form on Cloud Condensation Nuclei (CCN) and grow from the vapour phase. Eventually, as the cloud deepens and when drop size is sufficiently increased, the probability of collisions between droplets increases (Beard and Ochs III, 1993). Development of the droplet size distribution by condensation can be important if giant CCN (GCCN) and ultragiant aerosols (UGA) are present (e.g. Lasher-Trapp et al., 2001; Jensen and Lee, 2008). Turbulence has also been shown to enhance the rate of growth of raindrops (e.g. Ayala et al.,

2008; Grabowski and Wang, 2013) and also of cloud drops due to supersaturation fluctuations (e.g. Srivastava, 1989; Cooper, 1989; Prabhakaran et al., 2022).

The number density and size of the aerosol particles acting as CCN is critical for the initial properties of the droplet distribution at cloud base and hence the development of warm rain. Blyth et al. (2013) found that the development of warm

rain in RICO clouds could be explained by considering the aerosol particles alone. Measurements made at Ragged Point and the various aircraft showed that the was a significant variation in aerosol properties over the period of the project. Chazette et al. (2022) used lidar observations made on the French SAFIRE ATR42 research aircraft (ATR) and determined that the aerosols measured during EUREC[4]A are not surprisingly related to the synoptic situation. They found evidence of dust and



biomass burning aerosols originating from Africa. They also measured significant spatial and temporal variability in the concentration of aerosols. The reasons suggested for the variability were associated with the dynamical influence of the convection and the variability of the relative humidity.

The presence of Giant and Ultra Giant CCN leads to increased initial formation of larger particles, potentially enhancing the collision coalescence process and 'shortcutting' the warm rain process (Feingold et al., 1999). Production of these aerosol

over the ocean is potentially important in the cases discussed here (Woodcock et al., 1953).

Lonitz et al. (2015) investigated the role of aerosols and meteorology on the formation and development of precipitation in trade wind cumulus clouds over the Barbados Cloud Observatory (BCO) using a ground-based radar and lidar. These provided measurements of aerosols and clouds as well as vertical profiles of temperature and humidity. They determined that

the development of precipitation was influenced by the relative humidity as much as the amount of dust aerosols. They concluded that for the formation and development of precipitation it is very difficult to separate aerosol and meteorological effects.

Yamaguchi et al. (2019) performed large-eddy simulation modelling of clouds with no shear and found that an increase in

aerosol loading leads to deeper clouds, larger liquid water paths and surface rain, and a small decrease in cloud fraction. In other words, they showed that the well-known reduction in precipitation due to increased concentration of aerosols was buffered by this cloud deepening. They also found that the clustering of clouds protects them from entrainment and evaporation while the presence of wind shear caused enhanced evaporation.

One of the key goals of EUREC[4]A is to understand the importance of precipitation in these clouds. Accordingly, this paper addresses how the intensity and total amount of precipitation was influenced by the aerosol regimes associated with the different synoptic situations and by the clustering of the clouds. The analysis was made possible by the prevalence of subtropical cumulus cloud types in the region around Barbados, the varied aerosol and thermodynamic regimes and the excellent observation platforms and instruments in EUREC[4]A.

## 110   2 Overview of the measurement approach and measurement platform

The British Antarctic Survey Twin Otter (Fig. 1) is a high wing, twin engine, turbo prop aircraft. It has an operating range of about 1400 km and can cover an altitude up to 5000 m. The measurement approach during the project was to make in-situ measurements of boundary layer aerosol, above cloud top aerosol and the cloud layer itself. To meet these goals a typical flight consisted of a number of stepped straight and level flight segments throughout the depth of the cloud and measurements out

of cloud close to the surface, in the boundary layer and above cloud. An example of a typical flight pattern is shown in Figure 2. Fly-bys of the Ragged Point ground site were also performed for inter comparison of measurements made in flight and on the ground. A detailed overview of the EUREC[4]A science campaign can be found in Stevens et al. (2021).



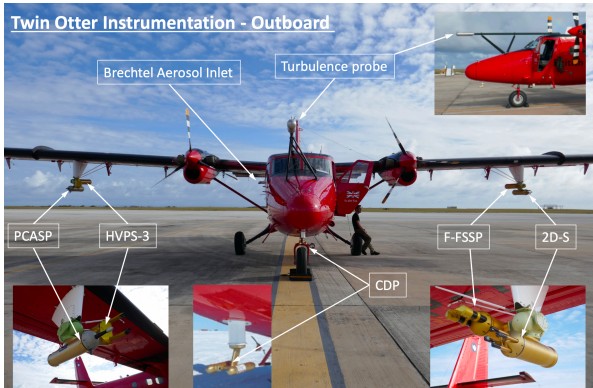

**Figure 1** The BAS twin otter aircraft with instrumentation and mounting positions labelled.


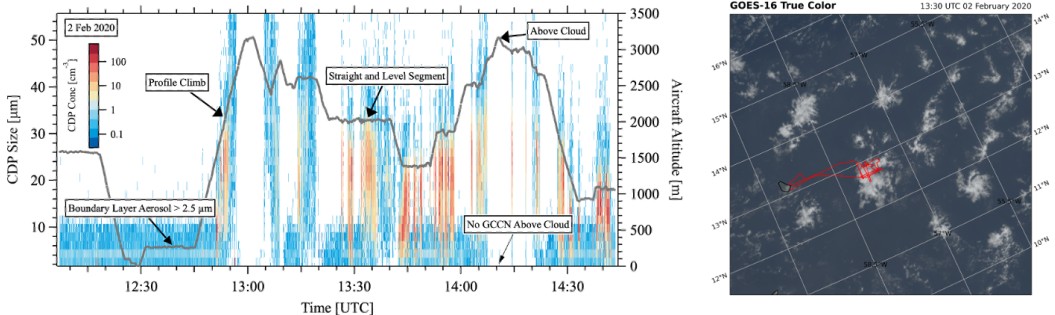

**Figure 2** Typical flight pattern (grey line) with profiles and straight and level manoeuvres to measure cloud and aerosol properties. Colour plot represents CDP size [μm] and concentration [cm$^{-3}$] as a function of flight time. GOES satellite image (right) for 2 Feb 2020 with flight track overlayed (red)


The Twin Otter was equipped with a suite of instruments to measure both the aerosol, cloud and precipitation particle size distributions. These included a (2D-S) Stereo Cloud Spectrometer, which provides two independent 2D optical imaging probes with 10 μm resolution images over the size range $10 < d_p < 1280$ μm (SPEC Inc) (Lawson et al., 2006); a Cloud Droplet Probe (Droplet Measurement Technologies, Boulder, USA) (Lance et al., 2010), that provided droplet size distributions over the

range $3 < d_p < 50$ μm; a Fast Forward Scattering Spectrometer (SPEC Inc) (Lawson et al., 2015) providing Particle By Particle (PBP) measurements over the range $1.5 < d_p < 50$ μm; a High Volume Precipitation Spectrometer version 3 (HVPS-3) that like the 2D-S provided shadow images, but at a resolution of 150 μm over the size range $150 < d_p < 19,200$ μm (SPEC Inc) (Lawson et al., 1998); a Passive Cavity Aerosol Spectrometer Probe (PCASP, DMT) measuring aerosol particles over the size range $0.1 < d_p < 3$ μm (Cai et al., 2013).


The ground-based site at Ragged Point utilised the mobile University of Manchester Laboratory to continuously sample from January 17$^{th}$ to February 22$^{nd}$ 2020 on the East Coast of Ragged Point (13.1638° N, 59.4329°W) with direct exposure to the Atlantic Ocean. Aerosols were sampled by pumped inlet (flow rate 1000 L min$^{-1}$) mounted on a 10 m tower and sub-sampled isokinetically by a suite of aerosol instruments covering sub-micron and super-micron size ranges, including a laser ablation

aerosol particle time of flight (LAAP-TOF) single particle mass spectrometer (Gemmayel 2016; Marsden et al., 2016) and a GRIMM optical particle counter (OPC) that measures aerosol between $0.3 < d_p < 20$ μm.



## 3 Meteorological Conditions and Aerosol Properties

During EUREC[4]A the Ragged Point ground site (Fig. 3) was used to measure atmospheric conditions including wind speed wind direction and temperature, together with information about the aerosol size distribution.

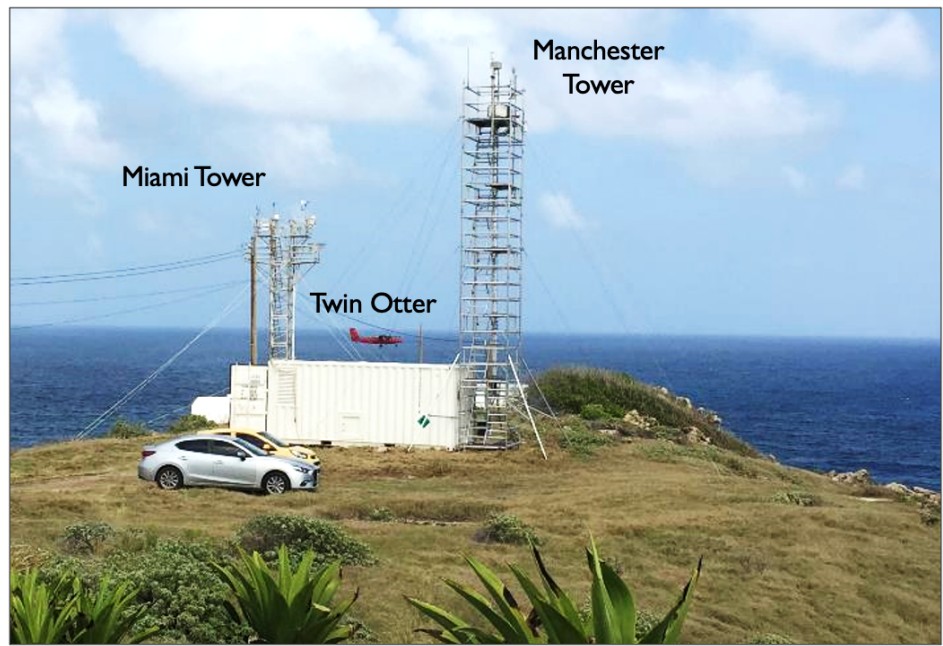


**Figure 3** Ragged Point ground site showing the Manchester tower constructed for making measurements, the Miami tower and the British Antarctic Survey (BAS) Twin Otter aircraft during a flypast of the site.

Wind direction during the campaign period was typically between South and East. Lower wind speeds were measured during
the first half of the campaign compared to the second at Ragged Point (RP) (Fig 4). The wind speeds measured during science flights on the Twin Otter were typically representative of those measured at RP – varying between about 5 and 15 m s$^{-1}$. Temperatures followed a diurnal cycle peaking around 25°C during the day.

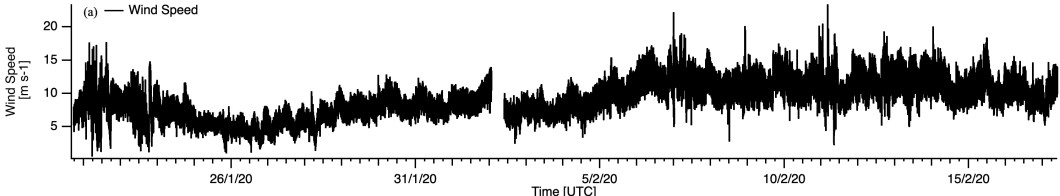

**Figure 4** Wind speed [m s$^{-1}$] measured at Ragged Point.

The campaign consisted of two distinct periods in which mean aerosol concentrations in the boundary layer were elevated (300 – 400 cm$^{-3}$ and 500 – 600 cm$^{-3}$ in the first and second period respectively) and periods where the concentrations of aerosol particles were lower (100-200 cm$^{-3}$). Figure 5 shows the median concentrations measured by the PCASP throughout the
campaign for boundary layer and above the inversion in the free troposphere (FT). The peak median concentrations in the two elevated periods are centred on 2 Jan and 9 Feb.



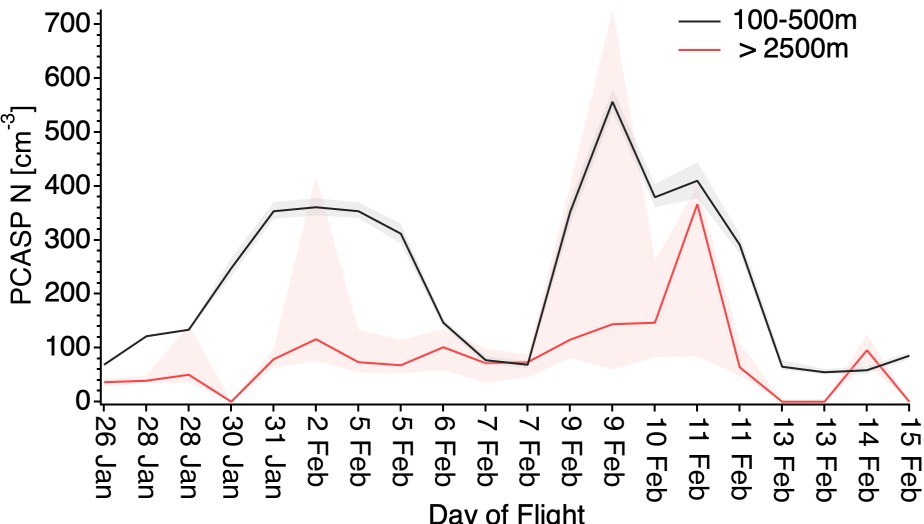

**Figure 5** Median PCASP number concentration in the boundary layer 100-500 m (black line) and free troposphere > 2500 m
(red line) as a function of flight date. Shaded areas represent the interquartile range.

Measurements in the FT, above the inversion level, showed typically lower aerosol concentrations for both the elevated periods
and the campaign period as a whole, however in some cases distinct layers of elevated aerosol concentrations were observed
close to or above the level of the inversion. Figure 6 shows the vertical profiles of aerosol measurements for a selection of
flights. The elevated layers were observed on 2 and 9 Feb (Fig. 6b, d and e) in the increased aerosol concentration periods.

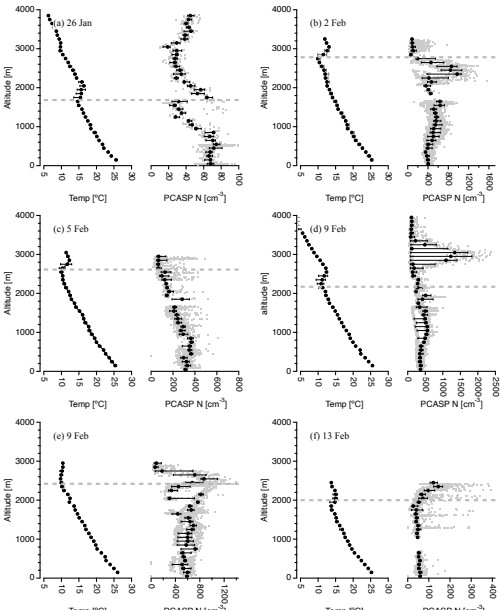

**Figure 6** Temperature and PCASP number concentration as a function of altitude for 26 Jan, 2, 5, 9 and 13 of Feb. The morning
and afternoon flights are both shown for 9 Feb. Black solid circles represent the median values, error bars represent 25[th] and
75[th] percentiles. Grey dots represent the 1Hz measurement of the PCASP instrument.




## 4. Ragged Point Ground Aerosol Observations

Measurements at the University of Manchester ground site at Ragged Point (RP) show clearly the two dominant elevated aerosol periods (Fig. 7) and that the first period (Fig. 8) was associated with dust aerosol and the second period with predominantly dust aerosol but with a signal for biomass burning.


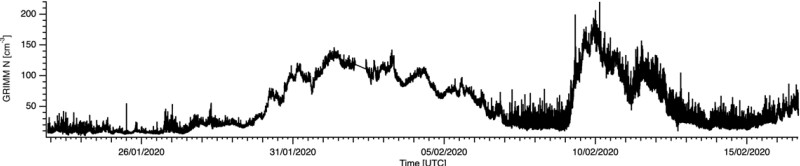

**Figure 7** Total number concentration from the GRIMM OPC during EUREC$^4$A.

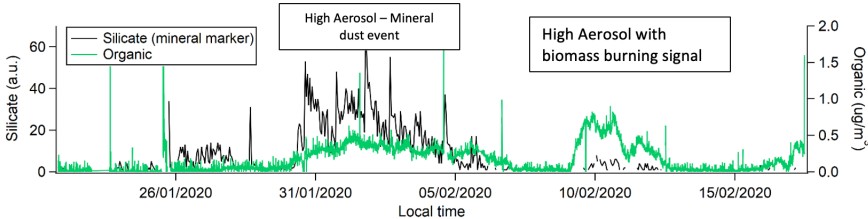

**Figure 8** Silicate (black line) and organic matter (green line) time series from the LAAPTOF.

The Twin Otter aircraft performed fly-by manoeuvres of RP during each of its flights and agreement between aerosol measurements made by the aircraft and at BCO (not shown) were found to be generally excellent across a range of different instruments and measurement techniques. The peak cloud base droplet number concentration in updrafts was found to be
strongly related ($r^2 = 0.89$) to the number of aerosol particles in the boundary layer as measured by the PCASP (Fig. 9).

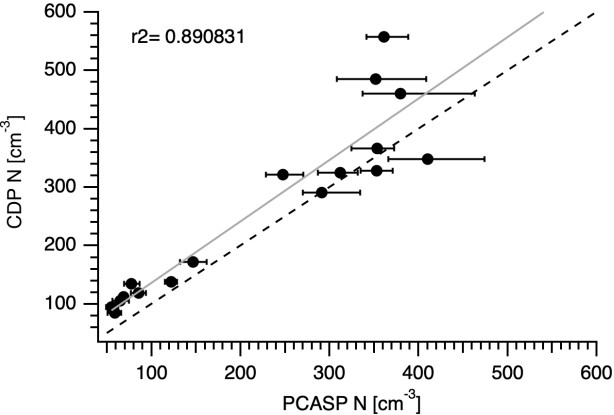

**Figure 9** Peak CDP concentration as a function of PCASP concentration in updraft regions close to cloud base. Black markers represent median values, whiskers represent 25[th] and 75[th] percentiles of the PCASP aerosol concentration. Dashed line 1:1,
solid grey line least squares fit with r2 of 0.89. PCASP measurements were made below cloud base and out of cloud.

This relationship was evident regardless of the aerosol period (elevated or relatively pristine). Composite particle size distributions from the PCASP and CDP during elevated and pristine conditions showed similar characteristics in terms of the



shapes of the distributions, with the changes in number concentration appearing to be attributed to increases or decreases in
number density across the whole spectrum. During some flights cloud droplet number concentration exceeded the PCASP
concentrations – this is attributed to strong updrafts and activation of CCN smaller than the PCASP detection limit.

## 5. Remote Sensing

Figure 10 shows a boxplot of the cloud top height (CTH) measured with the High Altitude and Long Range Research Aircraft
(HALO) cloud radar as part of the HALO Microwave Package (HAMP) (Mech et al., 2014). Examples of the measurement
approach is described in (Konow et al., 2021). There are two important points to make about the observations made during the
circles performed by the HALO aircraft. First, the characteristics of the clouds were often variable in space within the HALO
measurement circle and larger EUREC[4]A domain and with time over the 7-hour sampling period. Secondly, the number of
clouds with drops large enough to be sampled by the radar varied over the period of the project from as low as two clouds to
many clouds in fish type patterns.

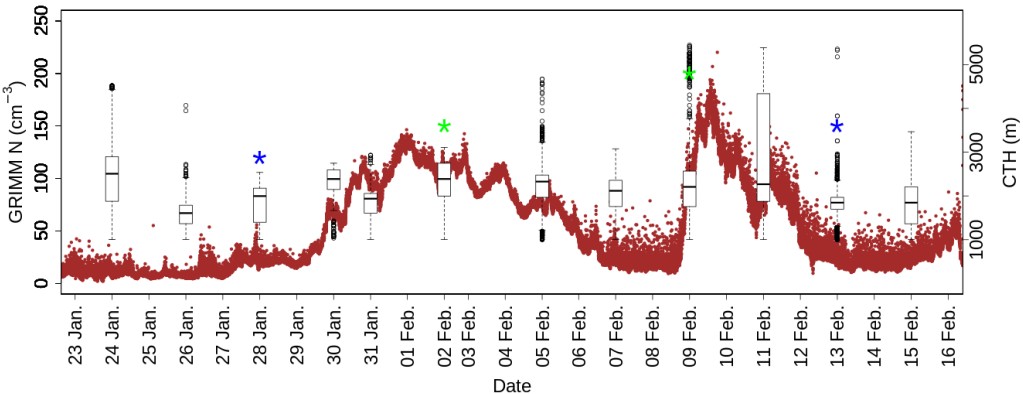

**Figure 10** (a) Boxplots of the cloud top height (m) from 23 January to 16 February 2020 determined from the HALO
Microwave Package (HAMP) cloud radar (35.5 GHz, Ka-band). The box represents the 25th (the first quartile, or Q1) to 75th
percentile (the third quartile, or Q3) range with a line inside the box for the median. The high (maximum) and low (minimum)
whiskers represent scores of 1.5 times of the interquartile range (IQR). Any outliers are represented by the black circles. Green
stars mark 2 high aerosol regime days and blue stars mark 2 low aerosol regime days. The 1Hz aerosol concentration data from
the GRIMM OPC is superimposed (red dots).


The median values of CTH were between 1500 and 2500 m for all cases. Figure 10 shows that generally the clouds that
developed in the low aerosol regime were shallower on average than those in the high aerosol regime (dust and/or biomass
burning particles). The aerosol concentration measured at Ragged Point by the GRIMM OPC shows the association between
the aerosol plume events and the higher CTHs. The inversion altitude (not shown) measured by the ATR research aircraft and
reported in Bony et al. (2022) also showed that the boundary layer height increases during the high aerosol regimes.

Figure 11 shows the HAMP cloud radar reflectivity for clouds in the low-aerosol regime on 28 January and 13 February, 2020
and in the high aerosol regime on the 2 and 9 February. The average cloud-top height on 2 and 9 Feb is greater than on 28 Jan
and 13 February. This can be seen in the altitude of the radar returns from HAMP. These differences are also shown in the





CTH boxplots in Figure 10. The blue stars represent the low aerosol regime cases and associated lower CTHs, while the green stars highlight the high aerosol regime cases and associated higher CTHs.

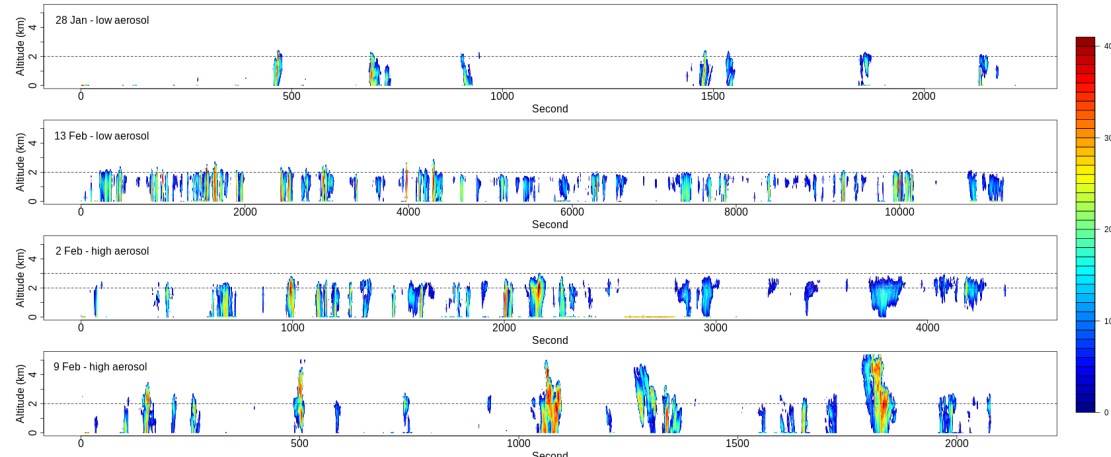

**Figure 11** HALO HAMP cloud radar reflectivity for all clouds with reflectivity echoes during the circles on 28 Jan, 2, 9 and 13 February, 2020.


Figures 10 and 11 suggest that the low and high-aerosol clouds developed in different air masses. The reason for the different CTHs being associated with the changing air masses is well described by Chazette et al. (2022). For example, the dust that is transported in a relatively deep layer across the Atlantic Ocean from Africa. Back trajectories (not shown) suggest that the air mass reaching the MBL was from the north on 28 January. The analysis performed by Chazette et al. (2022) suggests that on

2 February, there was a mix of dust aerosols from Mali and a mixture of biomass aerosols from the Western Sahara and South America. Chazette et al. (2022) showed that the relatively high concentration of aerosols extended to at least about 2.5 km which was a significantly deeper layer than on 28 January 2020. It is clear from the soundings released from Barbados (Fig. 12) that the cloud layer is significantly deeper on 2 February than on 28 January.

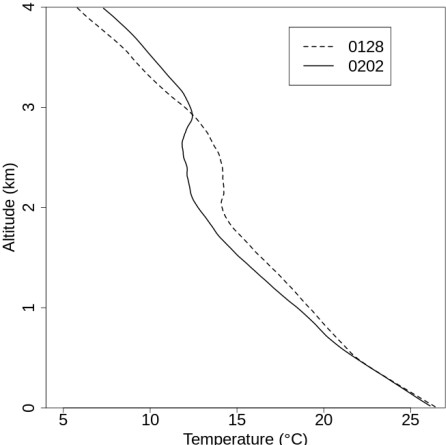


**Figure 12** Soundings taken at the BCO laboratory on 28 January, 2020 (a) during the low aerosol regime and 2 February, 2020 (b) during the high aerosol regime.



The cloud-top heights are more widely distributed in the high aerosol 9 February case than in the 2 February case, which can
be seen clearly in the HALO reflectivity (Fig. 11) and the boxplots of calculated cloud-top heights marked by green stars.
Unlike the 2 and 9 February cases, the CTH data on 11 February is positively skewed, i.e., there is a longer tail on the right
side of the distribution as the range between the third quartile and the median is about six times of the range between the
median and the first quartile. The CTHs are higher than the median before 3000 seconds in Figure 11 (c), but the CTHs are
more evenly distributed after 3000 seconds.


The CTHs on 13 February (marked by a blue star on Figure 10) are densely distributed near the median with the smallest Inter-
Quartile Range (IQR) of 270 m. A fish cloud system was the dominant type in the area of the HALO flight on the day and the
CTHs did not change much across the same cloud system (Fig. 11).

The main consequence suggested of the relationship between cloud depth and aerosol concentrations is that precipitation would
also be produced in the high aerosol clouds. In fact this can be seen from Figure 13. This shows that either the upper quartile,
or maximum reflectivity values is approximately 40 dBZ in the majority of cases that span across different aerosol
concentrations. The clouds sampled on 30 January were mainly sugar; there were only two clouds with rain reaching, or almost
reaching the ground. The maximum reflectivity values were slightly lower than 40 dBZ on the 26, 28 January and 2 and 9
February cases.

The 9 February case is curious because the clouds are so deep. The lowest scan made by the Poliarization Diversity Dopplar
Radar (PoldiRad) (not shown) measured reflectivity values of at least 40 dBZ on this day so it is likely that there was attenuation
in such deep clouds. Attenuation can be seen in the HAMP cloud radar reflectivity plot shown in Figure 11 particularly below
2000 m. Of course it is likely that other cases are also affected by attenuation, but not as severely since clouds were shallower
on other days.

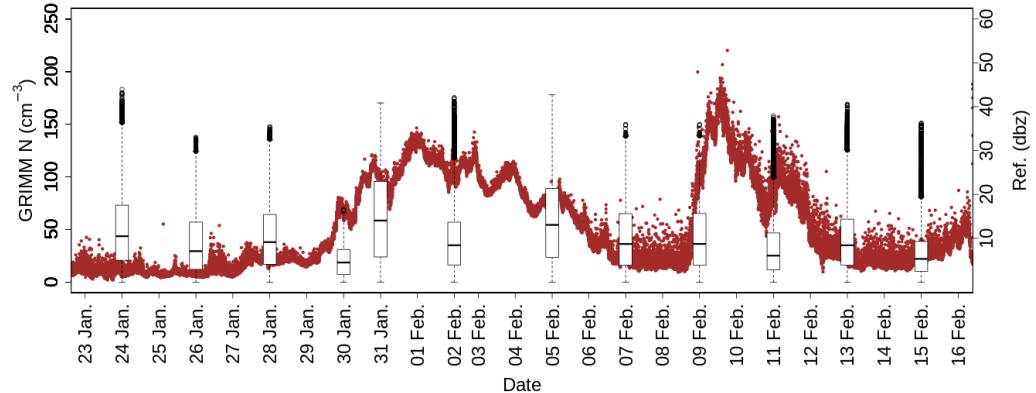

**Figure 13** Boxplots of the HALO HAMP radar reflectivity (dBZ) from 23 January to 16 February. The box represents the 25th
(the first quartile, or Q1) to 75th percentile (the third quartile, or Q3) range with a line inside the box for the median. The high
(maximum) and low (minimum) whiskers represent scores of 1.5 times of the interquartile range (IQR). Any outliers are shown
by the black circles. The 1Hz aerosol concentration data from the GRIMM OPC is superimposed (red markers).

**6. Cloud and precipitation properties – in-situ measurements**





Median profiles of the effective radius ($R_{eff}$) calculated from the CDP data are shown in Figure 14 for a selection of flights. Cloud bases were generally between 500-1000 m with greater variability in the depth of the cloud and height of the inversion (~ 1.5 km to ~ 3 km). Cloud Drop Number Concentration (CDNC) were observed to be significantly higher during periods of elevated aerosol. For example in the low aerosol regime on 28 January the median CDNC values (not shown) were ~ 60 cm$^{-3}$. During the flight on 9 February, which took place during the second period of elevated aerosol during the project, median

CDNC values were up to ~ 600 cm$^{-3}$.

Values of $R_{eff}$ measured by the CDP during cloud profiles had similar properties at cloud base. The initial $R_{eff}$ values were often around 5-10 µm for both the high aerosol and low aerosol regimes. The low and high aerosol cases that are presented in the HAMP radar reflectivity data (Fig. 11) are also shown in Figure 14. They all show a similarity in $R_{eff}$ at their respective

cloud bases, despite differences in CDNC and in the eventual development of clouds during each case. Figure 14 shows that with increasing cloud depth the rate of change in $R_{eff}$ was larger in the low aerosol regimes. For example in the low aerosol regime on 13 Feb the $R_{eff}$ increase from ~ 6 µm at an altitude of 800 m to ~ 18 µm at 1600 m. In the high aerosol regime on 5 Feb the $R_{eff}$ didn't reach similar values until 2800 m.



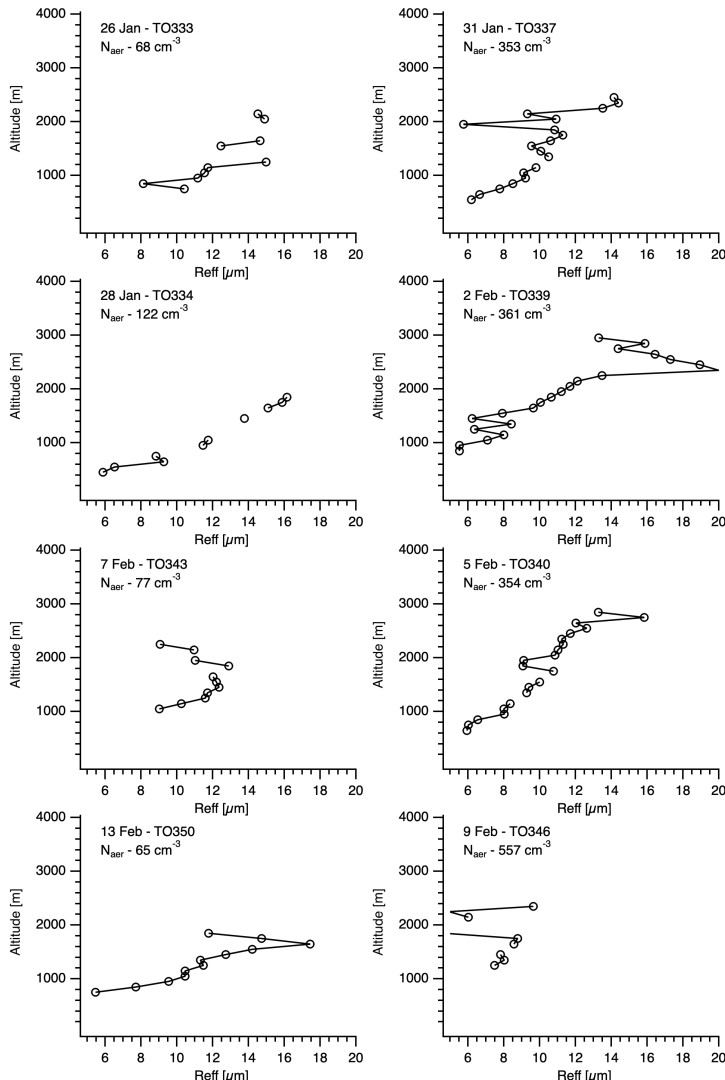

**Figure 14** vertical profiles of $R_{eff}$ for cases during the low aerosol regime (left panels) and the high aerosol regime (right panels). Markers represent median values of $R_{eff}$ binned as a function of altitude. The date of the flight, flight number and median boundary layer aerosol measured during the flight are labelled for each plot.

Figure 15 demonstrates the different $R_{eff}$ properties during the high and low aerosol regimes. The two groups show that the cases with lower aerosol rapidly increase in $R_{eff}$ when compared to cases with high aerosol loadings. However, the two groups achieve similar maximal values regardless of the initial conditions at cloud base and rate of increase with altitude.



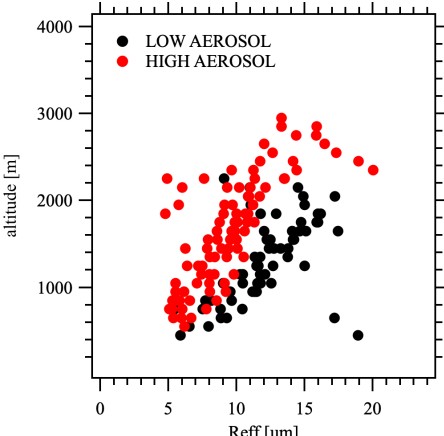

**Figure 15** $R_{eff}$ with altitude for low and high aerosol cases. Black markers represent low aerosol regimes. Red markers represent high aerosol regimes.

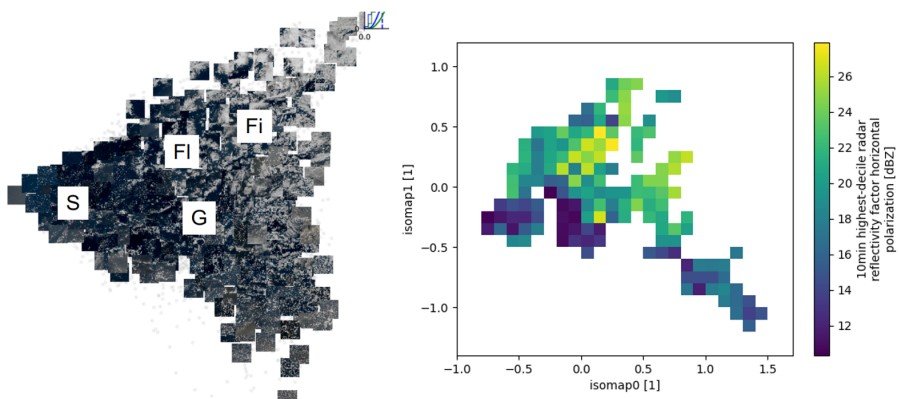

**Figure 16** a) Manifold of the 10 min average highest decile reflectivity measured by PoldiRad. a) Manifold showing satellite tiles with the annotation of sugar (S), gravel (G), flowers (Fl) and fish (Fi) clouds. b) Manifold of the 10 min average highest decile reflectivity measured by PoldiRad.

To investigate the extent to which intensity of rain in any individual cloud varies with mesoscale cloud organisation, the technique of Denby (2023) was used to characterise the mesoscale organisation in a 250km x 250km domain centred on the PoldiRad horizontal scanning swath (58.12W, 13.05N). The mesoscale organisation is characterised in Denby (2023) by using an unsupervised deep neural network, operating on RGB composite tiles created from GOES-16 observations, to produce so-called embedding vectors where similar cloud-scenes are placed close by in a high-dimensional embedding space. Using these embedding vectors the whole space of all possible kinds of organisation is mapped out into a manifold on which the PoldiRad observations are composited. To summarise the PoldiRad scans the median and highest-decile radar reflectivity across horizontal domain and over 10 minute intervals was computed (to match with the 10 min temporal resolution of GOES-16) excluding values outside of the -10dBz to 40dBz range (to exclude noisy values).

Figure 16 shows there is a clear shift in the radar reflectivity distribution with cloud organisation, with shallower scattered cloud (sugar) having lower reflectivity values, whereas the larger gravel, flower and fish clouds in general have larger values

of reflectivity.

Figure 17 shows the best fits derived from the scatter plots between the contiguous rain area and the maximum rain rate in that area using PoldiRad. The results indicate that the area of the cloud structure is an important factor for the development of the heaviest rain. Radtke et al. (2022) found a similar result, the drier scenes with more organised structures produced the

most rain. Figure 17 shows that the maximum rain rate for a fixed value of area varies from day to day, and the spread is more evident as the area increases. It is significant that the majority of the cases shown in Figure 17 were gravel clouds; only the 13 Feb case was a fish structure.

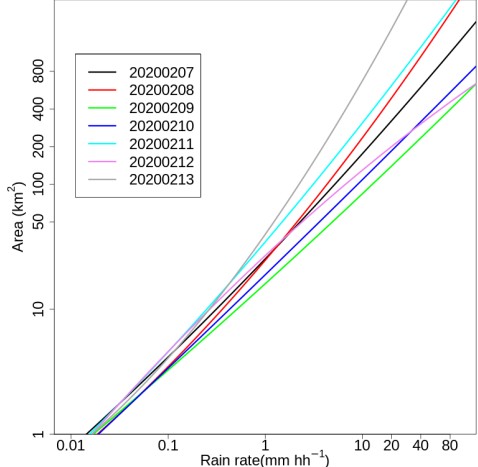

**Figure 17** Contiguous precipitating area as a function of maximum precipitation rate using the PoldiRad Radar data for best fit curves for all days from 7-13 February, 2022.

**7 Discussion and Conclusions**

The period of the field campaign consisted of three aerosol regimes with periods of low concentrations of aerosols, high concentrations of dust aerosols and high concentrations of a mixture of biomass-burning and dust aerosols. There were a few periods where the boundary layer contained low concentrations of aerosols that were mainly of local marine origin consisting of sulphate and sea-salt. The desert dust in the high-aerosol regime was found to have been advected across the Atlantic from the Sahara Desert and was generally observed to be in the atmospheric boundary layer. However, aerosol plumes containing

the highest concentrations were found on occasion in the free troposphere above the temperature inversion. When vigorous cloud penetrated the inversion there is evidence that some of this aerosol was entrained into the cloud along with dry air. Later in the project a mixture of biomass-burning aerosol and desert dust was observed in the boundary layer with concentrations of small particles much higher than the low-aerosol period and comparable to the early desert dust dominated period. The droplet spectra in the cloud as a function of height showed little systematic difference between the two high aerosol periods indicating

the aerosol was swell aged on its transit to the observation area with particles acting as efficient CCN.

The results suggest that the depth of the clouds was determined by the thermodynamic profiles, which in turn are influenced by the large-scale dynamics, but also by the local dynamics (e.g. collision of gust fronts). The elevated dust periods were



associated with a deeper boundary layer and consequently higher cloud tops than in the cases with low aerosol concentrations.
Hence, the maximum intensity of rain was similar for all the aerosol regimes.

While the maximum intensity of precipitation was not controlled by the aerosol loading, it was influenced by the area of cloud mass. Clouds that had somehow aggregated into a larger cloud mass generally produced heavier rain. This result was also found by Radtke et al. (2022). Larger masses of convective clouds were observed in the gravel-type clouds due to the collision
of gust fronts, and in the flower-type clouds due most likely to horizontal humidity convergence (e.g. Dauhut et al., 2023). The fish-type clouds were a large mass of convective clouds by definition. This conclusion is consistent with what is shown in Figure 16. There is little difference between the maximum precipitation intensities observed by PoldiRad in the fish flower and gravel type clouds.

Although the overall cloud structure appeared to be driven by the large scale dynamics, there was a significant aerosol influence on cloud microphysical properties during the pristine periods. Where it appears that rainfall is initiated lower down in the clouds (Fig. 15), possibly influenced by giant cloud condensation nuclei from the sea surface. In the deeper clouds with stronger vertical winds rain develops higher in the cloud due to diffusional growth of droplets followed by collision coalescence with little influence from giant CCN, any precipitation produced by giant CCN near cloud base is dominated by rainfall from higher
in the cloud.

**Data availability**

Facility for Airborne Atmospheric Measurements; Met Office; Natural Environment Research Council; Smith, M. (2004): Facility for Airborne Atmospheric Measurements (FAAM) flights. NCAS British Atmospheric Data Centre.
http://catalogue.ceda.ac.uk/uuid/affe775e8d8890a4556aec5bc4e0b45c

**Competing Interests**

The contact author has declared that none of the authors has any competing interests

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
