# Peer review of "The evolution of warm rain in trade-wind cumulus during EUREC4A"

_EGUsphere, 2024_

## Referee Comment (RC1)

**General Comments**

The manuscript titled "The evolution of warm rain in trade-wind cumulus during EUREC$^4$A" by Gary Lloyd et al. is based on aircraft and ground-based measurements of cloud and aerosol conditions during a wintertime field campaign conducted on or near Ragged Point, Barbados during 2020. It concludes that cloud depth is associated with thermodynamic profiles influenced by large-scale and local dynamics, the maximum intensity of precipitation is influenced by cloud mass area, and aerosols influence cloud microphysical properties in pristine aerosol regimes. The research is well within the scope of the journal. The findings are novel and worth publication.

Issues arise in the authors' writing style. The writing is, at times, convoluted and difficult to comprehend. The paper would be improved by simplifying sentence structures. Prepositional phrases, while useful, may complicate language and obscure the readers' understanding of the text. The authors also make statements about unreported data at several points throughout the text. Statements about data cannot be supported if data are not provided. I recommend that the authors include unreported data in the supporting information, reference existing literature in which the data they refer to is reported, or remove statements about unreported data entirely. I would recommend the editor reconsider the manuscript after major revisions by the authors.

**Specific Comments**

1. Line 9-10: Please rewrite this sentence to make it more concise.
2. Line 10: This is the first reference to EUREC$^4$A in the abstract. Please explain what the acronym stands for.
3. Line 12: This is the first reference to HALO in the text. Please explain what the acronym stands for.
4. Line 18-19: The sentence refers to "greatest intensities" but does not explain what the intensities are for. It does not follow from the previous sentence. Please explain what intensities you are referring to here.
5. Line 41: This is the first reference to EUREC$^4$A in the manuscript. Please explain what the acronym stands for.
6. Line 45-47: Please rewrite this sentence to make it more concise.
7. Line 63: This is the first reference to ICON in the text. Please explain what the acronym stands for.
8. Line 81: This is the first reference to SAFIRE in the text. Please explain what the acronym stands for.
9. Line 85-86: Please rewrite this sentence to make it more concise.
10. Line 105: The sentence refers to "these" clouds. Please specify which clouds "these" are. Shallow cumulus clouds?
11. Line 123: This is the first reference to CDP in the text. Please explain what the acronym stands for.
12. Line 123: This is the first reference to GOES in the text. Please explain what the acronym stands for.
13. Line 140: Please reference Marsden et al., 2016 in the bibliography.
14. Line 141: This is the first reference to GRIMM in the text. Please explain what the acronym stands for.
15. Line 145: This figure does not contribute much to the manuscript. Please remove.

16. Line 171: Would it be possible to organize the plots by high and low aerosol periods and adjust their PCASP N scales accordingly? For example, Jan 26, Feb 5, and Feb 13 are low aerosol conc days that could have the same scale and organized into the same column and Feb 2 and 9 (morning and afternoon) are high aerosol concentration days that could have the same scale and could be organized into a different column. This might make a greater point about the differences between high a low aerosol concentration conditions that you observed during the campaign. This is simply a suggestion to potentially improve reader comprehension.
17. Line 172-174: There is no mention of the subplot letters in the caption, though they are included in the figures. Please revise the caption such that each subplot has a corresponding subplot letter.
18. Line 172-174: It is not clear which Feb 9 subplot represents morning measurements and which represents afternoon measurements. Please specify
19. Line 187-189: This statement cannot be proven without data provided. Please include the data in the SI or manuscript, a reference to the data in pre-existing literature, or remove statement completely.
20. Line 211: Please explain what type of clouds "fish type patterns" refers to.
21. Line 221-223: Can you provide a value for this comparison? Perhaps average median CTH in low and high aerosol concentration conditions? Currently it is difficult to see this point from the figure alone.
22. Line 237-238: This is not a complete sentence. Are you referring to the type of air mass that might be associated with a particular CTH? Please revise.
23. Line 263: Please explain what type of clouds "sugar" refers to.
24. Line 267-269: This statement cannot be proven without data provided. Please include the data in the SI or manuscript, a reference to the data in pre-existing literature, or remove statement completely.
25. Line 283-285: This statement cannot be proven without data provided. Please include the data in the SI or manuscript, a reference to the data in pre-existing literature, or remove statement completely.
26. Line 324-325: Please provide descriptions for cloud types that have not yet been mentioned (gravel and flower).
27. Line 412: Cui et al., 2023 is not actually referenced in the manuscript. Please reference in the manuscript or remove from bibliography.
28. Line 425: Gemayel et al., 2016 is not actually referenced in the manuscript. Please reference in the manuscript or remove from bibliography.
29. Line 451: Marsden 2018 is not actually referenced in the manuscript. Please reference in the manuscript or remove from bibliography.
30. Line 466: Royer et al., 2023 is not actually referenced in the manuscript. Please reference in the manuscript or remove from bibliography.

Technical Corrections:
1. Line 9: Add comma after "In this paper"
2. Line 21: remove "the" before "daily weather"
3. Line 24: remove "names that describe"
4. Line 32: remove "determination of the"
5. Line 33: replace "to the" with "and"
6. Line 36: remove "even to this day"
7. Line 37: Add comma after "However"

8. Line 39: remove "the" before "cloud drops"
9. Line 39: Add comma before and after "for example"
10. Line 42: remove "in the case"
11. Line 42: Add comma after "However"
12. Line 51: remove "the" before cumulus clouds
13. Line 52: remove "need to"
14. Line 64: Add comma after "Interestingly"
15. Line 65-66: remove "the reasons were that"
16. Line 69: Add comma after precipitation
17. Line 70: "Cloud Condensation Nuclei" should not be capitalized
18. Line 81: replace "the" with "there" before "was a significant variation"
19. Line 83: remove "surprisingly"
20. Line 88: "Giant and Ultra Giant" should not be capitalized
21. Line 106: remove "total"
22. Line 126: remove "both"
23. Line 126: Add a comma after "cloud"
24. Line 126: remove "particle"
25. Line 130: "Particle By Particle" should not be capitalized
26. Line 143: Add a comma after "EUREC$^4$A"
27. Line 143: Add a comma after "wind speed"
28. Line 144: Remove comma after "temperature"
29. Line 149: Remove "period"
30. Line 151: Remove "science"
31. Line 158: Add a comma before "respectively"
32. Line 67-69: This is a run-on sentence. Please separate into 2 sentences. I suggest the first end at "whole" and the next sentence start with "However,".
33. Line 188: Remove "to be generally excellent"
34. Line 234: Abbreviate "February" to "Feb" to be consistent with the rest of the manuscript
35. Line 282: remove "observed to be significantly"
36. Line 283: Add a comma after "For example"
37. Line 287: Please explain the abbreviation "$R_{eff}$" in the manuscript.
38. Line 319: Add a comma after "summarise"
39. Line 329: Remove comma after "result"
40. Line 356: Add a comma after "Later"
41. Line 357: Remove "in the project"
42. Line 360: Remove "swell"
43. Bibliography: Please make sure the bibliography is in agreement with journal guidelines.

---

## Author Comment (AC1)

General Comments

The manuscript titled "The evolution of warm rain in trade-wind cumulus during EUREC$^4$A" by Gary Lloyd et al. is based on aircraft and ground-based measurements of cloud and aerosol conditions during a wintertime field campaign conducted on or near Ragged Point, Barbados during 2020. It concludes that cloud depth is associated with thermodynamic profiles influenced by large-scale and local dynamics, the maximum intensity of precipitation is influenced by cloud mass area, and aerosols influence cloud microphysical properties in pristine aerosol regimes. The research is well within the scope of the journal. The findings are novel and worth publication.

Issues arise in the authors' writing style. The writing is, at times, convoluted and difficult to comprehend. The paper would be improved by simplifying sentence structures. Prepositional phrases, while useful, may complicate language and obscure the readers' understanding of the text. The authors also make statements about unreported data at several points throughout the text. Statements about data cannot be supported if data are not provided. I recommend that the authors include unreported data in the supporting information, reference existing literature in which the data they refer to is reported, or remove statements about unreported data entirely. I would recommend the editor reconsider the manuscript after major revisions by the authors.

Specific Comments

1. Line 9-10: Please rewrite this sentence to make it more concise.
   *AC: We have modified this sentence.*
2. Line 10: This is the first reference to EUREC$^4$A in the abstract. Please explain what the acronym stands for.
   *AC: We have defined the acronym.*
3. Line 12: This is the first reference to HALO in the text. Please explain what the acronym stands for.
   *AC: We have defined the acronym.*
4. Line 18-19: The sentence refers to "greatest intensities" but does not explain what the intensities are for. It does not follow from the previous sentence. Please explain what intensities you are referring to here.
   *AC: We have re-written this sentence to make it clearer, but the absolute values and relationship we found is described in the main text and in figure 17.*
5. Line 41: This is the first reference to EUREC$^4$A in the manuscript. Please explain what the acronym stands for.
   *AC: We have defined the acronym.*
6. Line 45-47: Please rewrite this sentence to make it more concise.
   *AC: We have altered this sentence.*
7. Line 63: This is the first reference to ICON in the text. Please explain what the acronym stands for.
   *AC: We have defined the acronym.*
8. Line 81: This is the first reference to SAFIRE in the text. Please explain what the acronym stands for.
   *AC: We have defined the acronym.*

9. Line 85-86: Please rewrite this sentence to make it more concise.
   *AC: We have re-written this sentence to make it more concise.*
10. Line 105: The sentence refers to "these" clouds. Please specify which clouds "these" are. Shallow cumulus clouds?
    *AC: This has been properly defined as shallow trade wind cumulus clouds.*
11. Line 123: This is the first reference to CDP in the text. Please explain what the acronym stands for.
    *AC: Acronym defined.*
12. Line 123: This is the first reference to GOES in the text. Please explain what the acronym stands for.
    *AC: Acronym defined.*
13. Line 140: Please reference Marsden et al., 2016 in the bibliography.
    *AC: We have added this reference.*
14. Line 141: This is the first reference to GRIMM in the text. Please explain what the acronym stands for.
    *AC: We believe GRIMM is actually the surname of the creator.*
15. Line 145: This figure does not contribute much to the manuscript. Please remove.
    *AC: We would like to keep this figure as a useful record of the overall instrument setup at the ground site.*
16. Line 171: Would it be possible to organize the plots by high and low aerosol periods and adjust their PCASP N scales accordingly? For example, Jan 26, Feb 5, and Feb 13 are low aerosol conc days that could have the same scale and organized into the same column and Feb 2 and 9 (morning and afternoon) are high aerosol concentration days that could have the same scale and could be organized into a different column. This might make a greater point about the differences between high a low aerosol concentration conditions that you observed during the campaign. This is simply a suggestion to potentially improve reader comprehension.
    *AC: These have been reorganized and the x axis fixed for low and high cases for comparison.*
17. Line 172-174: There is no mention of the subplot letters in the caption, though they are included in the figures. Please revise the caption such that each subplot has a corresponding subplot letter.
    *AC: This has now been defined in the caption.*
18. Line 172-174: It is not clear which Feb 9 subplot represents morning measurements and which represents afternoon measurements. Please specify
    *AC: This is now stated on the figure and in the caption.*
19. Line 187-189: This statement cannot be proven without data provided. Please include the data in the SI or manuscript, a reference to the data in pre-existing literature, or remove statement completely.
    *AC: We have removed the statement and the subsequent figure as the paragraph did not fit well with the flow of the paper. This change may help with the convoluted nature of the paper.*
20. Line 211: Please explain what type of clouds "fish type patterns" refers to.
    *AC: These are defined visually due to their fish like structure on visible satellite imagery. We have referenced the original paper that classified the different cloud types.*

21. Line 221-223: Can you provide a value for this comparison? Perhaps average median CTH in low and high aerosol concentration conditions? Currently it is difficult to see this point from the figure alone.
    *AC: We have calculated the difference in CTH between the two aerosol regimes and stated this in the paper. The difference is ~ 300 m.*
22. Line 237-238: This is not a complete sentence. Are you referring to the type of air mass that might be associated with a particular CTH? Please revise.
    *AC: We have removed this sentence.*
23. Line 263: Please explain what type of clouds "sugar" refers to.
    *AC: We have defined this at the end of the paragraph. Sugar is described in Stevens et al. (2022) as a dusting of very fine-scale clouds with small vertical extension and little evidence of self-organisation (by cold pools or gust fronts).*
24. Line 267-269: This statement cannot be proven without data provided. Please include the data in the SI or manuscript, a reference to the data in pre-existing literature, or remove statement completely.
    *AC: We have removed this statement.*
25. Line 283-285: This statement cannot be proven without data provided. Please include the data in the SI or manuscript, a reference to the data in pre-existing literature, or remove statement completely.
    *AC: We have removed this statement.*
26. Line 324-325: Please provide descriptions for cloud types that have not yet been mentioned (gravel and flower).
    *AC: We have now included a proper description in the introduction of these types of clouds rather than simply referencing the original paper that defined them. We hope this helps the reader to better picture the varying structures observed during the campaign.*
27. Line 412: Cui et al., 2023 is not actually referenced in the manuscript. Please reference in the manuscript or remove from bibliography.
    *AC: Reference removed.*
28. Line 425: Gemayel et al., 2016 is not actually referenced in the manuscript. Please reference in the manuscript or remove from bibliography.
    *AC: This is now corrected and included in the manuscript.*
29. Line 451: Marsden 2018 is not actually referenced in the manuscript. Please reference in the manuscript or remove from bibliography.
    *AC: This is now corrected and included in the manuscript.*
30. Line 466: Royer et al., 2023 is not actually referenced in the manuscript. Please reference in the manuscript or remove from bibliography.
    *AC: This has been removed from the bibliography.*

Technical Corrections:
1. Line 9: Add comma after "In this paper"
   *AC: We have added this.*
2. Line 21: remove "the" before "daily weather"
   *AC: We have removed 'the'.*
3. Line 24: remove "names that describe"
   *AC: This has been removed.*

4. Line 32: remove "determination of the"
   *AC: This has been removed.*
5. Line 33: replace "to the" with "and"
   *AC: This has been replaced.*
6. Line 36: remove "even to this day"
   *AC: This has been removed.*
7. Line 37: Add comma after "However"
   *AC: Comma added.*
8. Line 39: remove "the" before "cloud drops"
   *AC: 'the' removed.*
9. Line 39: Add comma before and after "for example"
   *AC: Comma added.*
10. Line 42: remove "in the case"
    *AC: Removed.*
11. Line 42: Add comma after "However"
    *AC: Comma added.*
12. Line 51: remove "the" before cumulus clouds
    *AC: Removed.*
13. Line 52: remove "need to"
    *AC: Removed.*
14. Line 64: Add comma after "Interestingly"
    *AC: Added.*
15. Line 65-66: remove "the reasons were that"
    *AC: Removed.*
16. Line 69: Add comma after precipitation
    *AC: Comma added.*
17. Line 70: "Cloud Condensation Nuclei" should not be capitalized
    *AC: Capitalization removed.*
18. Line 81: replace "the" with "there" before "was a significant variation"
    *AC: Replace.*
19. Line 83: remove "surprisingly"
    *AC: Removed.*
20. Line 88: "Giant and Ultra Giant" should not be capitalized
    *AC: Capitalization removed.*
21. Line 106: remove "total"
    *AC: Removed.*
22. Line 126: remove "both"
    *AC: Removed.*
23. Line 126: Add a comma after "cloud"
    *AC: Added.*
24. Line 126: remove "particle"
    *AC: Removed.*
25. Line 130: "Particle By Particle" should not be capitalized
    *AC: Capitalization removed.*
26. Line 143: Add a comma after "EUREC$^4$A"

*AC: Comma added.*

27. Line 143: Add a comma after "wind speed"
    *AC:  Comma added.*
28. Line 144: Remove comma after "temperature"
    *AC: Comma removed.*
29. Line 149: Remove "period"
    *AC: Removed.*
30. Line 151: Remove "science"
    *AC: Removed.*
31. Line 158: Add a comma before "respectively"
    *AC: Comma added.*
32. Line 67-69: This is a run-on sentence. Please separate into 2 sentences. I suggest the first end at "whole" and the next sentence start with "However,".
    *AC: This has been separated into two sentences.*
33. Line 188: Remove "to be generally excellent"
    *AC: Sentence removed as covered in specific comments.*
34. Line 234: Abbreviate "February" to "Feb" to be consistent with the rest of the manuscript
    *AC: February changed to Feb.*
35. Line 282: remove "observed to be significantly"
    *AC: Removed.*
36. Line 283: Add a comma after "For example"
    *AC: Comma added.*
37. Line 287: Please explain the abbreviation "$R_{eff}$" in the manuscript.
    *AC: This has now been defined in the manuscript at the start of section 6.*
38. Line 319: Add a comma after "summarise"
    *AC: Comma added.*
39. Line 329: Remove comma after "result"
    *AC: Comma Removed.*
40. Line 356: Add a comma after "Later"
    *AC: Comma removed.*
41. Line 357: Remove "in the project"
    *AC: This has been removed.*
42. Line 360: Remove "swell"
    *AC: Removed.*
43. Bibliography: Please make sure the bibliography is in agreement with journal guidelines.
    *AC: Formatting checked.*

---

## Author Comment (AC2)

**The evolution of warm rain in trade-wind cumulus during EUREC[4]A**

This paper characterizes the thermodynamic and microphysical variabilities within shallow trade cumulus clouds near Barbados, primarily using aircraft observations from the EUREC[4]A campaign in 2020. The study analyzes shallow cumulus precipitation structures in relation to aerosol loadings and cloud mesoscale structure. Key findings show that maximum rain intensity is associated with cloud aggregation rather than aerosol loading, though aerosols impact cloud depths. The in-depth characterizations of aerosol-cloud microphysical and macrophysical processes using various in-situ and remote sensing platforms are valuable for future modeling studies and the broader shallow cloud research community.

However, the paper currently lacks structure and logical flow. The paragraphs do not transition smoothly, and the objective of the paper is not clear until much later in the text. Some major revisions are necessary to improve clarity and readability. Below are some major and minor comments that suggest more organization and some clarification regarding key results. With this, I am recommending a major revision.

Major comments:

1. The premise of the paper should be clarified early in the introduction. The interest in examining the effects of aerosol on precipitation intensity in different conditions and clustering should be stated upfront. Reorganize the introduction into three broad sections: Research topics and science interests, previous works and their highlighted results, and current research objectives. In general, ensure each paragraph and sections end with statements leading to the next paragraphs and sections. This will help improve the readability and comprehension.

   *AC: We have restructured the introduction and made additions based on the recommendations. We now state the aims early in the introduction and have clarified the goals of the research more clearly. We also made changes to improve the readability of the introduction.*

2. Include all instrumentation and measurement details in section 2. Include the wind and temperature measurements description that is in section 3 right now. Elaborate the Twin Otter and HALO flight details, days, number of flights, duration, distance between HALO and TO flights etc. Write the abbreviations of all the instruments, campaign, satellite etc. used in this study. Currently, many of them are just mentioned as acronyms.

   *AC: We merged section 3 with section 2 to improve the structure of the measurement descriptions and combine the wind and temperature information with all the other descriptions. The flight details have been stated including the date range and number of flights. We also refer to existing published research describing the EUREC4A measurement domains and the various instrument platforms involved in the project and their proximity to each other. We have checked to make sure abbreviations for all of the instruments are given where appropriate.*

3. Section 2 includes all the microphysical probes on board the TO. However, the only ones used in this paper are PCASP and CDP. Could the analysis include the precipitation rates and precipitation drop size distribution evolution from clean to polluted cases using the 2DS and HVPS samples? This could confirm the linkage of aerosol loading and precipitation intensity using independent platforms.

   *AC: We have completed new analysis to calculate the rain rates in the low and high aerosol regimes using the HVPS drop size distribution. This is described in the in-situ measurement*

*section. We found that the aerosol regime doesn't appear to control the precipitation intensity. The main finding is that the mesoscale cloud structure does influence the precipitation intensity, with the larger mesoscale cloud structure (flowers and fish) associated with the greatest rain rates.*

4.  Expand the discussion to link findings to tie it up with introduction. State the limitations of the approach and suggest the scope and need for future research.

    *AC: We have expanded the discussion and added limitations and some ideas for potential modelling work to build on the observations from this paper.*

Minor comments:

- Line 41: Write the full form of EUREC4A here since you mention it for the first time. State that the manuscript is based on the EUREC4A datasets.

    *AC: We have defined the acronym and made sure it is consistent throughout the manuscript.*

- Line 50: "*The motivation for previous projects and the work presented in this paper focus on the importance of the cumulus clouds in the trade wind region around Barbados and the difficulty they pose for models that need to parameterise clouds and atmospheric properties in such environments.*" Clarify how your motivation and objective add to previous studies, the differences in methodologies, timescales, study areas, if any.

    *AC: We have added a statement describing the way EUREC$^4$A adds to the previous studies in the final paragraph of the introduction.*

- Line 143: The wind and temperature measurements are described in section 3. It should be part of section 2 along with all the other instrumentations.

    *AC: The structure of paper has been modified extensively, with all the measurement information now described in section 2 along with the other instruments.*

- Figure 8: The dust aerosols described throughout the manuscript is referred to as 'silicate (mineral marker)' in the legend. Adding 'dust aerosol' in the legend will be helpful. Similarly, add 'biomass' in the legend as well beside 'organic' for consistency

    *AC: We have included these in the legend.*

- Figure 8: Change 'LAAPTOF' in the figure description to 'LAAP-TOF' for consistency.

    *AC: This has been changed.*

- Line 186: "*The Twin Otter aircraft performed fly-by manoeuvres of RP during each of its flights and agreement between aerosol measurements made by the aircraft and at BCO (not shown) were found to be generally excellent across a range of different instruments and measurement techniques*". Could the list of instruments at BCO used for this verification be listed in section 2?

    *AC: As we do not show the inter-comparison we have now decided to remove this statement.*

- Figure 9: The percentiles of the CDP concentration could also be included on the y-axis. Mention if each point is representative of each flight.

  *AC: This has now been removed as part of the revisions to improve the structure of the paper.*

- The HALO flights and instrumentations including HAMP used in this study should also be included in section 2.

  *AC: This manuscript has been edited to make sure all of the relevant information about the instruments and measurement platforms is contained within section 2.*

- Figure 10a: There is only one panel, so 'a' should be removed.

  *AC: This has been removed.*

- Line 237: Rephrase "For example, the dust that is transported in a relatively deep layer across the Atlantic Ocean from Africa."

  *AC: This has been rephrased.*

- Line 249: Is the higher CTH spread for 9 Feb compared to 2 Feb case linked to the presence of biomass and dust?

  *AC: It's unclear whether this is linked to the presence of different aerosol types. It's likely to have been driven by the thermodynamic profile of the atmosphere in the region being conducive in some places to deep convection.*

- Use CTH instead of cloud-top heights consistently.

  *AC: This has been checked and CTH used throughout.*

- Line 264: Could the reflectivity at the lowest radar range gate be used to see if aerosol concentration ($N_a$) still does not correlate with reflectivity? Additionally, could the 2DS and HVPS observations mentioned in section 2 be used to compute rain rates, and then correlated with $N_a$ to re-confirm this result

  *AC: There is attenuation to consider for the lowest levels in particular. In any case the reflectivity values within the cloud are a better representation of the warm rain process due to evaporation below cloud base.*

  Figure 13: What is the altitude of radar reflectivity shown in the figure? In Figure 11, the radar reflectivity closest to the surface seems to have higher reflectivity on 2 and 9 February compared to 28 Jan and 13 Feb. If so, then could the cloud base reflectivity (and hence rain intensity) be correlated with $N_a$? A scatter plot showing the cloud base reflectivity/rain rate/CTH vs $N_a$ would be more intuitive (instead of time series) for emphasizing the key points here.

  *AC: The statistics were calculated using all the radar reflectivity values at all altitudes for a particular day. The exception is that the large values near the surface where there were no clouds above were excluded. The reviewer makes a good suggestion about a scatter plot. However, we opted for the analysis shown mainly because of presentation, but also because: a) there are some relatively high reflectivity values close to the surface on 13 Feb; b) there is attenuation to consider; and c) the statistics are better if all reflectivity values are included.*

*We included the first part of this answer in the figure caption.*

- For figures 10 and 13, include the correlation coefficients between $N_a$ and CTH and reflectivity for low and high $N_a$ It is hard to follow the boxplot median lines as a function of GRIMM N.

  *AC: To make the correlation between the $N_a$ and CTH clearer we have split the regimes and calculated the median CTH for the high and low aerosol periods. We found the CTH to be ~ 300 m greater in the high aerosol regime. This has been included in the paper.*

- Figure 14: Could the panels be arranged by date or $N_a$ for better readability?

  *AC: The panels have been split by $N_a$.*

- Line 300: "*However, the two groups achieve similar maximal values regardless of the initial conditions at cloud base and rate of increase with altitude.*" This line contradicts the paragraph at line 286. This earlier paragraph says that the Reff is similar at cloud base but the rate of increase in Reff is higher in low $N_a$ But line 300 conveys that regardless of initial (Reff) conditions at cloud base, the rate of increase is the same. Could this be clarified?

  *AC: We have clarified that the reason the cases achieve similar Reff is due to the higher CTHs in the higher aerosol regimes.*

- Paragraphs after line 313 do not fit into the section 6 headline. Use new section for this.

  *AC: This has now been combined with the remote sensing section.*

- Tie the Figure 16 and 17 results with the previous results. For more context, the essential features (e.g., reflectivity, spatial width, rain rates) of the mesoscale structures (fish, flower, gravel, sugar) should be defined in the introduction. Later, in the results sections the dates featuring each of these structures should be indicated both in text and figures.

  *AC: We have now defined in detail the different mesoscale cloud structures (fish, flowers, gravel and sugar) in the introduction. The different mesoscale organisation is indicated in Figure 16 and with the manifold highlighting the continuum of cloud scenes that have variation between the more strictly defined cloud groups.*

- What is the significance of the x- and y- axis in Figure 16? Is the shape of the map indicative of anything? Some clarity would be helpful for readers not acquainted with neural networking.

  *AC: We attempt to describe this difficult concept in the manuscript, the axes represent the embedding vectors where cloud scenes identified as similar in nature by the neural network are mapped onto the manifold using these vectors.*

- Figure 17 and paragraph at line 327: How does this figure tie in with the previous sections? Among all the days shown in the figure, 13 Feb with fish clouds seems to have the least rain rate for a given area. However, in the previous paragraph fish clouds are linked with higher reflectivity which should be a proxy for higher rain intensity. Clarification will be helpful.

  *AC: There will be some variation in rain rates even within a particular cloud group. The lower rain rates on 13 Feb in the fish clouds that were presented in Figure 17 also tie in with those presented in new rain rate figure X, with the in-situ measurement observing higher rain*

*rates in the flowers on 7 Feb vs the 13 Feb fish case. This is also consistent with the data presented in Figure 17.*

- Why are the other days described in the rest of the paper (26,28,31 Jan, 2,5 Feb) not shown in Figure 17?

  *AC: Unfortunately due to delays the Poldirad radar did not arrive until later in the project so we only present the data we had available.*